# FDG-PET/CT, a Promising Exam for Detecting High-Risk Myeloma Patients?

**DOI:** 10.3390/cancers12061384

**Published:** 2020-05-28

**Authors:** Anne-Victoire Michaud-Robert, Bastien Jamet, Clément Bailly, Thomas Carlier, Philippe Moreau, Cyrille Touzeau, Mickael Bourgeois, Françoise Kraeber-Bodere, Caroline Bodet-Milin

**Affiliations:** 1Nuclear Medicine Department, Nantes University Hospital, 44093 Nantes, France; annevictoire.michaud@chu-nantes.fr (A.-V.M.-R.); bastien.jamet@chu-nantes.fr (B.J.); clement.bailly@chu-nantes.fr (C.B.); thomas.carlier@chu-nantes.fr (T.C.); mickael.bourgeois@univ-nantes.fr (M.B.); francoise.bodere@chu-nantes.fr (F.K.-B.); 2CRCINA, INSERM, CNRS, Angers University, Nantes University, 44093 Nantes, France; philippe.moreau@chu-nantes.fr (P.M.); cyrille.touzeau@chu-nantes.fr (C.T.); 3Hematology Department, Nantes University Hospital, 44093 Nantes, France; 4Nuclear Medicine Department, (Institut de Cancérologie de l’Ouest) ICO René Gauducheau, 44805 Saint Herblain, France

**Keywords:** multiple myeloma, FDG-PET/CT, prognostic value, smouldering multiple myeloma, high-risk patients

## Abstract

Multiple myeloma (MM) is a haematological neoplasm characterized by a clonal proliferation of malignant plasma cells in the bone marrow. MM is associated with high morbidity and mortality and variable survival, which can be very short for some patients but over 10 years for others. These differences in survival are explained by intra- and inter-tumoral heterogeneity and demonstrate the potential benefits of adapting the treatment course for high-risk patients with a poorer prognosis. Indeed, identification of these high-risk patients is necessary and is based on the identification of high-risk biomarkers including clinical variables, genomics and imaging results. Positron emission tomography combined with computed tomography using 18F-deoxyfluoroglucose (FDG-PET/CT) is a reliable technique for the initial staging of patients with symptomatic multiple myeloma (MM), and has been included in the IMWG (International Myeloma Working Group) recommendations in 2019. According to clinical studies, FDG-PET/CT characteristics could be used to define high-risk patients at initial diagnosis of symptomatic MM. The goal of this review is to demonstrate the prognostic value of FDG-PET in symptomatic MM patients, particularly in identifying high-risk patients, and thus, to best adapt therapeutic management in the future.

## 1. Introduction

Positron emission tomography combined with computed tomography using 18F-deoxyfluoroglucose (FDG-PET/CT) is a non-invasive functional imaging modality of the whole body. It is a reliable technique for the initial staging of patients with a multiple myeloma (MM) and has been included in the 2019 International Myeloma Working Group (IMWG) recommendations as a feasible imaging strategy for the initial workup of newly diagnosed MM [1]. FDG-PET/CT detects myeloma related lesions with excellent sensitivity and specificity [2] with the advantage of carrying out both bone and extra-bone exploration in a single examination.

MM is characterized by the clonal proliferation of plasma cells that is almost always preceded by an initial monoclonal gammopathy of undetermined significance (MGUS). This then develops into asymptomatic smouldering multiple myeloma (SMM), which constitutes an intermediate clinical stage between MGUS and MM. The definition of MM [3] was revised in 2014 by the IMWG [4].

MM is associated with high morbidity and mortality [5] and variable survival, being very short for some patients but over 10 years from diagnosis for others. These differences in survival are explained by intra- and inter-tumoral heterogeneity and demonstrate the potential benefits of adapting the treatment course for high-risk patients (with a poor prognosis). Indeed, identification of these high-risk patients is necessary and is based on the identification of high-risk biomarkers including clinical variables, genomics and imaging results. The past decade has seen considerable advances in developing risk classifiers based on cytogenetics and gene expression profiling [6], but spatial heterogeneity can limit the sensitivity of these tests because they are based on cells obtained from a single bone marrow biopsy. Rasche et al. [6] found that high-risk genomic alterations can be present in one focal lesion yet completely absent in other locations, highlighting the fact that bone marrow biopsies obtained from a single site do not necessarily reflect the entire bone marrow. The same team found that spatial differences were seen primarily in patients with large focal lesions, suggesting that imaging can be used to identify patients with extensive heterogeneity based on the size of the lesion(s) [7]. Combined with the results of others studies [2,8,9], several FDG-PET/CT characteristics could be defined as possible high-risk biomarkers and could be used to defined high-risk patients at initial diagnosis of symptomatic MM.

On the other hand, FDG-PET/CT also benefits from its ability to identify at an early stage those SMM patients at high risk of progression to symptomatic MM [10] and who require initiation of appropriate treatment.

The goal of this manuscript is to clarify the prognostic value of FDG-PET/CT at initial diagnosis of SMM, symptomatic MM, solitary plasmacytoma (SP) and at relapse, in order to identify high-risk patients and to optimize therapeutic management.

## 2. Prognostic Value of FDG-PET/CT Negativity at Baseline in Symptomatic Multiple Myeloma Patients

The performance of FDG-PET/CT in detecting MM is excellent, with a sensitivity of around 90% for the detection of myeloma lesions, and a specificity varying from 70 to 100% in several studies [11,12], with a greater sensitivity than whole-body conventional radiography [13] and comparable sensitivity with pelvic-spinal MRI [2,14,15].

FDG-PET/CT is negative in approximately 10–20% of patients. This result, considered as a false negative for the detection of the disease, should be considered above all for its prognostic value.

Indeed, Rasche et al. showed that FDG-PET/CT could be considered as ineffective for ≈11% of patients due to low expression of hexokinase 2 (which catalyses the first step of glycolysis), and is responsible for a false negative diagnosis in this group of patients [16]. In another study with 90 newly diagnosed MM patients receiving novel agents during induction therapy, Abe et al. showed that low hexokinase 2 expression associated with a false negative PET-FDG/CT was associated with relatively better prognosis for patients with newly diagnosed MM [17].

Recently, Moreau et al. presented the results of the CassioPET study at the 2019 American Society of Haematology congress, a companion study to the Cassiopeia prospective trial that demonstrates the superiority of combined daratumumab-VTD (Velcade-Thalidomide-Dexametasone) treatment over VTD alone in young patients before autologous transplantation [18]. The CassioPET study showed the prognostic value of a negative pre-therapeutic FDG-PET/CT. Indeed, patients with negative baseline FDG-PET/CT (20%) had significantly better progression-free survival (PFS) compared to patients with positive baseline FDG-PET/CT (HR = 0.42, *p* = 0.0365) [19]. This observation is very important and suggests that a negative FDG-PET/CT may allow one to orient the risk based on therapeutic strategy.

## 3. Prognostic Value of FDG-PET/CT Abnormalities at Baseline

### 3.1. Symptomatic Multiple Myeloma

FDG-PET/CT allows detection of different abnormalities in MM patient bone marrow. The bone marrow abnormalities are focal lesion (FL), para-medullary disease (PMD) and diffuse bone marrow involvement with variable glucose uptake, resulting in a variable standard uptake value (SUV) [11,16,20]. FDG-PET/CT also allows the detection of extra-medullary disease (EMD) in less than 10% of patients at diagnosis.

FL is defined as focal uptake above the surrounding background noise on two successive sections with or without osteolysis opposite on the CT image (excluding uptake due to a fracture, arthrosis or benign bone disease) (Figure 1). PML is defined as soft tissue invasion with contiguous bone involvement (Figure 2), diffuse medullary involvement as a homogenous diffuse uptake of the pelvic-spinal skeleton that may extend to the peripheral skeleton of greater intensity than the liver and EMD as tissue invasion without contiguous bone involvement (Figure 3).

Four large prospective studies have demonstrated the important prognostic impact of FDG-PET results at baseline. Indeed, four PET biomarkers have been identified in these cohorts of patients receiving different therapeutic protocols: the number of FLs, the SUVmax of FLs, the presence of EMD and the presence of PMD (Table 1).

Firstly, in a large cohort of 239 MM patients treated using a Total Therapy 3 strategy, a first line treatment in a double autograft program [8], Bartel et al. showed that the only diagnostic imaging modality between FDG-PET/CT and MRI significantly associated with unfavourable prognosis for both overall survival (OS) and event-free survival (EFS) was FDG-PET/CT when the number of FLs was >3. The number of FLs on the baseline MRI (7 or more) affected only EFS.

In a large series of 192 MM patients treated using first line thalidomide-dexamethasone induction therapy and double autologous stem cell transplantation (ASCT), the Bologna group confirmed the pejorative prognostic impact of more than 3 FLs on PFS at 4 years, as well as a SUVmax of FL > 4.2 and the presence of EMD. A SUVmax of FL > 4.2 and the presence of EMD were also associated with a shorter OS [9]. This publication by the Bologna group is the first that demonstrates the prognostic interest in the SUVmax for this indication. While many publications have already demonstrated the limitations of this quantitative parameter, it is useful for its easy use in routine practice [22].

The prognostic value of EMD on PFS and OS was confirmed by the prospective multicentric French IMAJEM study that included 134 newly diagnosed MM patients randomized to the lenalidomide-bortezomib-dexamethasone (RDV) arm alone or RVD followed by ASCT [2], but no prognostic value was reported for the SUV max or the number of FLs.

More recently, the CassioPET prospective study examined a large cohort of 268 symptomatic MM patients and confirmed the prognostic value of EMD detected at baseline by FDG-PET/CT in 7.8% of patients (*p* = 0.0341) [19]. Interestingly, this prospective study was the first to also report the prognostic value of PMD detected by FDG-PET/CT in 17.5% of patients at baseline (*p* = 0.0002).

Several other studies have attempted to find or confirm the potential impact of these PET prognostic biomarkers at baseline. In a smaller study including patients with MM (55 patients) and SP (6 patients), Haznedar et al. [23] demonstrated a correlation between the most intense EMD FDG-uptake and both osteo-medullary uptake (*p* = 0.027) and the International Staging System (ISS) score (*p* = 0.048). The bone marrow SUVmax was correlated with the ISS score (*p* = 0.013). The 44 patients with positive FDG-PET/CT had a shorter five-year survival than the 11 patients with negative FDG-PET/CT, all of whom were alive after five years (*p* = 0.01). Multivariate analysis revealed only the EMD with the highest uptake had a prognostic value on OS (*p* = 0.03). More recently, in a retrospective study including 228 MM patients, Abe et al. found that the presence of more than three FLs in the appendicular skeleton on baseline FDG-PET/CT was an independent predictor of poor PFS and OS in patients with newly diagnosed MM (*p* < 0.001). On the other hand, the number of FLs and EMD were not discriminating in terms of prognosis in this study [21].

Several studies have also evaluated the prognostic value of baseline volume-based FDG-PET parameters. These metrics, such as metabolic tumor volume (MTV) and total lesion glycolysis (TLG), appeared as promising tools by quantifying functional disease burden in MM [24]. In a recent study examining 192 patients with MM, baseline TLG greater than 620 g and MTV greater than 210 cm^3^ were associated with poor PFS and OS after adjusting for baseline myeloma variables. Combined with the 70-gene expression profiling (GEP70) risk score, a TLG greater than 205 g identified a high-risk subgroup, and divided ISS stage II patients into two subgroups with similar outcomes to ISS stage I and ISS stage III [25]. In a retrospective study of 47 patients with MM stage IIIA, Fonti et al. confirmed the prognostic impact of MTV. PFS and OS were significantly better for patients with MTV ≤ 39.4 mL (*p* = 0.0004 and *p* = 0.0001 respectively) as compared to those having an MTV higher than the cut off when adjusted by multivariate analysis. However, the use of MTV is limited in clinical practice and can only be considered after standardisation of its determination method [26].

New techniques are currently being developed to determine the prognostic value of FDG-PET/CT for the diagnosis of MM. In 2019, Morvan et al. used random forest survival (RSF) for the first time on PET imaging for the progression prediction of MM patients at baseline from a database of 66 patients who were part of the IMAJEM study. They found that combining textural features (radiomics) with clinical and biological data could improve the prediction of PFS [27]. These results need to be confirmed, in particular by carrying out new prospective studies and the use of a validation cohort.

The new IMWG guidelines on imaging monoclonal plasma cell disorders published in 2019 [28], recommended that whole-body CT is the first-choice imaging technique to identify and assess the extent of osteolytic lesions, but FDG-PET/CT can be used instead of whole-body CT or whole-body MRI, only if the CT part of FDG-PET/CT fulfils the criteria of a whole-body CT diagnosis (Figure 4).

### 3.2. Smouldering Multiple Myeloma 

SMM is defined by the presence of at least 10% of monoclonal plasma cells in the bone marrow, as well as by the presence of monoclonal proteins in the serum or urine, and in some cases both, without associated CRAB criteria (hypercalcemia, renal injury, anemia, bone disease) [4]. SMM is a heterogeneous classification that includes patients whose progression to symptomatic MM is very slow (several years) and patients whose progression to symptomatic MM is very rapid (less than 2 years, high-risk SMM). Consequently, the definition of symptomatic MM was revised in 2014 by the IMWG. This integrated new pejorative biomarkers including medullary plasmacytosis ≥60%, a serum free light chain ratio ≥100 and more than 1 focal bone lesion (≥5 mm in size) detected by magnetic resonance imaging (MRI) in addition to the usual CRAB criteria, with the aim of not delaying the initiation of treatment for patients classified as high risk SMM [4].

Whilst the 2014 IMWG recommendations indicate that the presence of one or more FL only with osteolysis detected by FDG-PET/CT is pathologic, a prospective study from 2009 suggested that positive foci with or without osteolysis could precede morphological abnormalities. In SMM, a positive FDG-PET/CT defined by the presence of FL without underlying osteolytic lesions, is associated with a rapid progression to symptomatic MM (Figure 5). Indeed, in a cohort of 122 SMM patients, Siontis et al. [29] found that the probability of progression to MM within 2 years with positive FDG-PET/CT was 75% vs. 30% for patients with a negative FDG-PET/CT, without therapy. In another prospective study of 120 SMM patients, all without evidence of underlying osteolysis, the Bologna group showed the same results: the progression to symptomatic MM at 2 years for FDG-PET positive patients was 58% vs. 33% for patients with a negative FDG-PET/CT [10]. However, as these results were published after the most recent IMWG guidelines for the definition of MM, they have not been considered as a prognostic biomarker. In the new IMWG guidelines on imaging in monoclonal plasma cell disorders published in 2019 [28], FDG-PET/CT can be used instead of whole-body CT or whole-body MRI (if the MRI procedure is not feasible), only if the CT part of FDG-PET/CT fulfils the criteria of a whole-body CT diagnosis (Figure 4) [30].

### 3.3. Solitary Plasmacytoma

The SP is defined as a localized proliferation of malignant plasma cells in either bone or soft tissue, without evidence of MM and whose prognosis is marked by a high-risk of transformation to MM.

FDG-PET/CT allows detection of additional bone or soft tissue lesions with greater sensitivity (98%) and specificity (99%) compared to MRI [31]. In a series of 43 SP patients, Fouquet et al. showed that the presence of at least two hypermetabolic lesions on FDG-PET/CT was predictive of a rapid progression to MM [32]. Using multivariate analysis, they showed that abnormal initial involved serum-free light chain (*p* = 0.008) and FDG-PET/CT (*p* = 0.032) were independently associated with a shorter MM transformation time.

Although FDG-PET/CT appears to have better sensitivity for the detection of additional bone lesions, whole-body MRI is still recommended for the initial diagnosis of a solitary bone plasmacytoma.

FDG-PET/CT is recommended only if MRI is not feasible. However, if a diagnosis of extra-medullary solitary plasmacytoma is made, FDG-PET/CT is recommended as the first imaging modality (Figure 6) [28].

## 4. Prognostic Value of FDG-PET/CT Abnormality at Relapse

The prognostic value of FDG-PET/CT was also evaluated for patients with MM suspected of relapse. In a series of 37 MM patients suspected of relapse after ASCT, Lapa et al. [33] showed that the absence of hyper metabolic foci on FDG-PET/CT was a favourable prognostic factor for both time to progression (TTP) and OS (*p* < 0.01). The presence of more than 10 FLs was correlated with a shorter time to progression and OS, as was the glucose uptake intensity and presence of EMD. The result of the FDG-PET/CT led to a change in the treatment regimen in 30% of cases.

More recently, in a retrospective study of 40 confirmed relapsed patients, Jamet et al. [34] described that the presence of at least six FLs in the appendicular skeleton was an independent pejorative prognostic factor for both PFS and OS (*p* = 0.01 and *p* = 0.04, respectively). Moreover, a high SUVmax (>15.9) was an independent negative prognostic factor for PFS (*p* = 0.047), as was a high TLG of the hottest lesion (>98.1) (*p* = 0.04). 

## 5. Perspective

Several studies have shown the pejorative prognostic impact of several pathologic features including the number [8,9] and size [7] of FLs, the maximal tracer uptake measured using SUVmax [9], the presence of EMD [2,8,9] and the presence of PMD [19], which are associated with a high-risk disease. By combining these FDG-PET/CT imaging results with clinical variables and genomics data, it would be possible to develop new guidelines in the future for risk stratification based on patient data obtained from large prospective trials, with the aim of early identification of high-risk patients at diagnosis and adapting the treatment to the severity of the disease. Moreover, tumor heterogeneity appears to have a potential prognostic impact that remains to be confirmed, several studies using machine-learning techniques suggest that radiomics could be useful in managing patients [27].

However, ^18^F-deoxyglucose is not a specific tracer in PET/CT and inflammatory cells are likely to generate false positive results. New radiotracers such as ^18^F-fludarabine have been compared to ^18^F-deoxyglucose in pre-clinical studies and shown to specifically localize MM lesions [35]. Another interesting perspective is the chemokine receptor 4 (CXCR4) molecule. CXCR4 is expressed at high levels on the surface of monoclonal plasma cells and is activated by its ligand CXCL12. The CXCR4/CXCL12 axis is an essential pathway in normal hematopoietic stem cell niche regulation in the bone marrow. However, CXCR4/CXCL12 signaling also plays a critical role in proliferation, invasion, dissemination, and drug resistance in MM. In addition to its role in homing, CXCR4 also affects the mobilization of MM cells and their escape from the BM, which correlates with metastatic spread to distant organs [36]. Aberrant expression of CXCR4 is associated with osteoclastogenesis and tumor growth in MM through its cross-talk with various important cellular signaling pathways [37]. It was also observed that persistent chemo-resistant MRD plasma cell clones in MM express high levels of CXCR4, while abrogation of the CXCR4/CXCL12 pathway can deregulate BM colonization by hematopoietic cells. Recently, a radio-labelled CXCR4-ligand for PET imaging called ^68^Ga-pentixafor has been developed by the Wurzburg group. Dosimetry and proof-of-concept for visualization of CXCR4-expression using ^68^Ga-pentixafor-PET has been demonstrated in MM patients in a pilot study allowing non-invasive detection of CXCR4 expression in 23/35 MM patients [38]. This preliminary data also suggest that CXCR4 expression represents a negative prognostic factor. They also demonstrated the potential of CXCR4 as a theranostic target in MM patients in a pilot study confirming the feasibility of CXCR4-directed radionuclide therapy using ^177^Lu or ^90^Y-CXCR4 ligand (pentixather) as a novel treatment approach for MM [39]. Interestingly, compared to ^18^FDG, ^68^Ga-pentixafor also seems to be a promising imaging tool for assessment of other related blood cancers including MALT lymphoma [40], Waldenström macroglobulinemia, lymphoplasmacytic lymphoma [41] and primary central nervous system lymphoma [42]. Furthemore, the use of 11C-methionine (MET) in MM also seems to have a promising future. Lapa et al. prospectively compared the sensitivity of ^18^FDG and MET for the detection of myeloma lesions in patients evaluated at baseline (*n* = 11) or in relapse (*n* = 32). PET-MET was able to detect LF and EMD in 6 and 2 more patients respectively than FDG-PET and also found more LF and EMD in 28 and 6 cases respectively. Thus, they demonstrated that MET uptake was correlated with bone marrow involvement, β2-microglobulin and free light chain levels, and appears to be a more accurate market of tumor burden compared to ^18^FDG [43]. Another recent study confirmed the excellent sensitivity of MET for the assessment of MM tumor burden compared to ^18^FDG [44]. Nevertheless, the prognostic value of this tracer need further investigation particularly in comparison with ^18^FDG, which is currently the strongest marker in this pathology.

In addition to the use of PET/CT, the emergence of PET/MRI could be an interesting way for exploring MM and others plasma cell dyscrasias, with the ability to image the whole body using both, PET and MRI by combining the excellent performances of these two exams, notably in high-risk patients, for initial staging, to assess treatment response and at relapse [45].

## 6. Conclusions

FDG-PET/CT is an imaging technique with a strong ability to detect medullary and extra-medullary disease at the initial diagnosis of MM with a pejorative prognostic value of several features, including probably radiomics, associated with high-risk disease. Whilst patient prognostic stratification is currently based on laboratory tests and genomic abnormalities, functional imaging like FDG-PET/CT may be an important additional tool for patient stratification, particularly for defining a high-risk disease patient group, and thus, to best adapt therapeutic management in the future. 

## Figures and Tables

**Figure 1 cancers-12-01384-f001:**
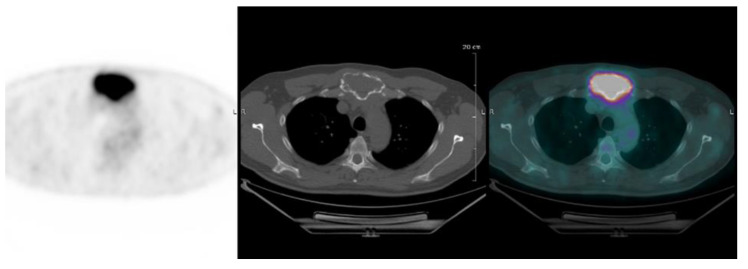
Focal hypermetabolism of an osteolytic lesion in the sternum.

**Figure 2 cancers-12-01384-f002:**
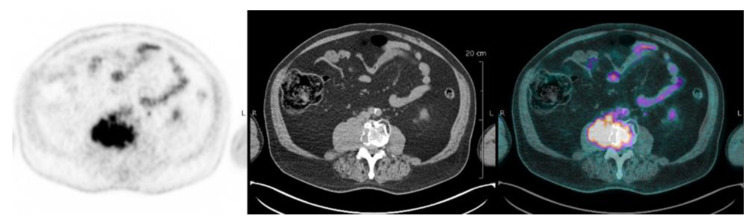
Hypermetabolic lytic lesion of L3, contiguously invading adjacent soft tissues.

**Figure 3 cancers-12-01384-f003:**
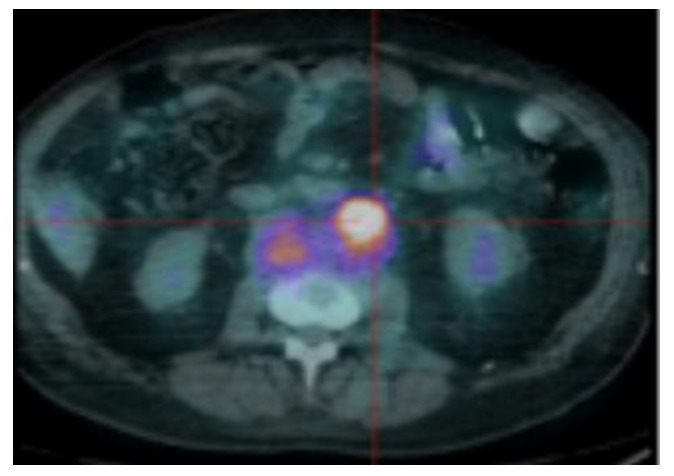
Extra-medullary pre-aortic hypermetabolism corresponding to an EMD.

**Figure 4 cancers-12-01384-f004:**
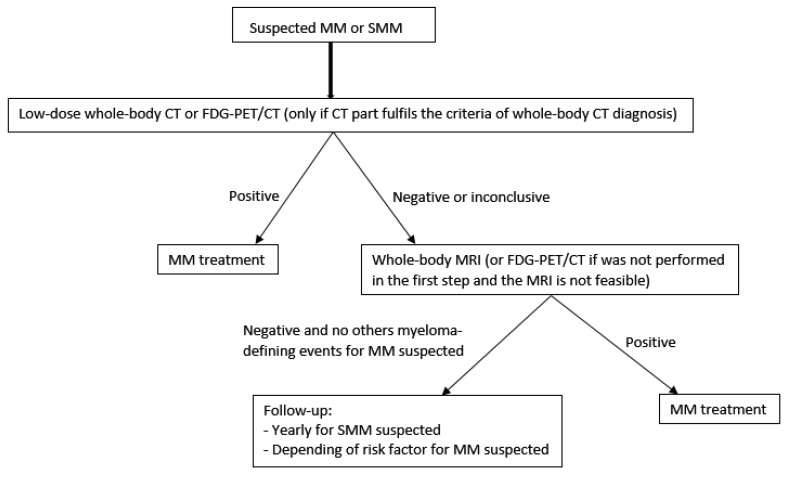
Imaging algorithm for MM and SMM suspected. CT: computerized tomography; MRI: magnetic resonance imaging; MM; multiple myeloma; SMM: smoldering multiple myeloma. Based on “Hillengass J, Usmani S, Rajkumar SV, Durie BGM, Mateos M-V, Lonial S, et al. International myeloma working group consensus recommendations on imaging in monoclonal plasma cell disorders. The Lancet Oncology. 2019” [28].

**Figure 5 cancers-12-01384-f005:**
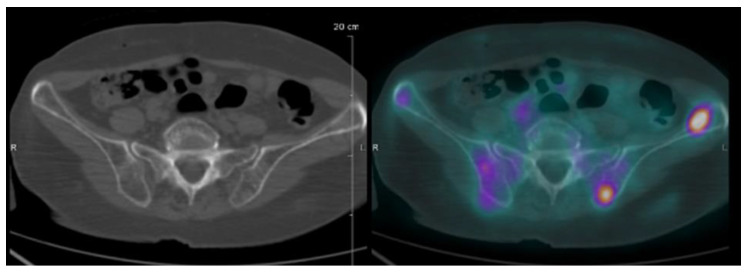
FDG-PET/CT images of an IgG SMM patient. Two FDG-positive focal lesions of the iliac left bone were identified without osteolytic lesion on the CT scan. A biopsy of the accessible area showed a focal accumulation of clonal plasma cells, upgrading the patient to symptomatic MM requiring treatment, and demonstrated the prognostic value of FDG-PET in a case of SMM.

**Figure 6 cancers-12-01384-f006:**
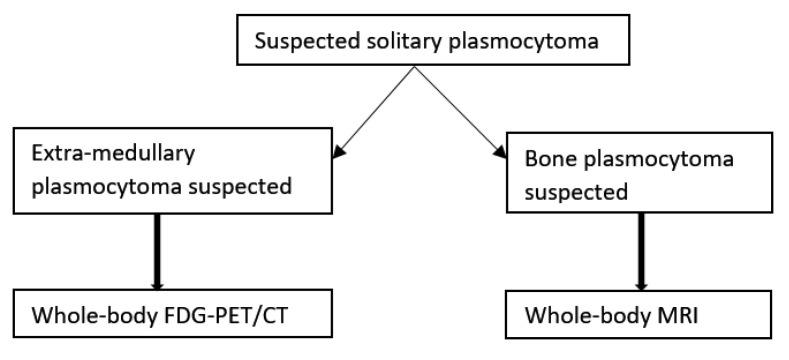
Imaging algorithm for patients with solitary plasmocytoma suspected. MRI: magnetic resonance imaging. Based on “Hillengass J, Usmani S, Rajkumar SV, Durie BGM, Mateos M-V, Lonial S, et al. International myeloma working group consensus recommendations on imaging in monoclonal plasma cell disorders. The Lancet Oncology. 2019” [28].

**Table 1 cancers-12-01384-t001:** Definition and prognosis value of FDG-PET/CT parameters in symptomatic MM.

Interpretation of PET Based on Prognostic Biomarkers			
Lesions	Definition	Prognostic Biomarker	Reference
Focal lesion	Foci of uptake above the surrounding background noise on two successive sections with or without osteolysis on the CT image	Suggested as pejorative prognostic biomarker using cut off 3.	Bartel et al., 2009 [8]
			Zamagni et al., 2011 [9]
			Abe et al., 2019 [21]
EMD	Tissue invasion without contiguous bone involvement.	Presence of EMD suggested as pejorative prognostic biomarker	Zamagni et al., 2011 [9]
			Moreau et al., 2017 [2]
			Moreau et al., 2019 [19]
PMD	Soft tissue invasion with contiguous bone involvement.	Presence of PMD suggested as pejorative prognostic biomarker	Moreau et al., 2019 [19]
Diffuse medullary involvement	Homogenous diffuse uptake of the pelvic-spinal-peripheral skeleton higher than the liver background.	Prognostic value currently not demonstrated	
FL SUVmax	Maximal SUVmax of bone focal lesions	Suggested as pejorative prognostic biomarker using cut off of 4.2	Zamagni et al., 2011 [9]
FDG-PET/CT abnormality	Presence of Focal lesion(s) and/or EMD lesion(s) and/or PMD lesion(s) and/or diffuse medullary involvement.	Absence of FDG-PET/CT abnormality considered as a favourable prognostic	Rasche et al., 2017 [16]
			Moreau et al., 2019 [19]
			Abe et al., 2019 [17]

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
