# Peer review of "FDG-PET/CT, a Promising Exam for Detecting High-Risk Myeloma Patients?"

_cancers, 2020, doi:10.3390/cancers12061384_

Round 1

Reviewer 1 Report

This a very well-written review on the role of PET and particularly [18F]FDG-PET in high-risk myeloma. Essentially, I have two suggestions:

  1. The part on other tracers should be expanded a bit. While I do agree that CXCR4 tracers could ultimately have more potential than other tracers (also due to the option of targeted endoradiotherapy), the reports for [11C]Methionine are at least equally promising – in fact, in terms of detection rates, Methionine might even considered the best tracer currently available for MM (the Lapa group actually favors it over Pentixafor as far as I know); there are several articles in Theranostics, Clin Nucl Med, and also a very recent article in Cancers, comparing it to FDG.
  2. More references are needed on the role of CXCR4; there are currently many free-standing sentences without references. It should also be mentioned that Pentixafor has been successfully applied in related blood cancers, i.e., PCNSL, MALT lymphoma, CLL and Morbus Waldenström.

Author Response

FDG-PET/CT, A PROMISING EXAM FOR DETECTING HIGH-RISK MYELOMA PATIENTS?

REPLY TO REVIEWER #1

We would like to thank the Reviewer #1 for all the valuable comments and for the careful review of the manuscript, which has helped to improve our work. In response to the reviewer’s concerns, we have added some references in the text. Please find below a detailed point-by-point response to all comments (reviewers’ comments in black, our replies in blue, suggested text change in red).

 1. The part on other tracers should be expanded a bit. While I do agree that CXCR4 tracers could ultimately have more potential than other tracers (also due to the option of targeted endoradiotherapy), the reports for [11C]Methionine are at least equally promising – in fact, in terms of detection rates, Methionine might even considered the best tracer currently available for MM (the Lapa group actually favors it over Pentixafor as far as I know); there are several articles in Theranostics, Clin Nucl Med, and also a very recent article in Cancers, comparing it to FDG.

Response:  We thank the reviewer for raising this important observation and in response we have developed a paragraph in “Perspective” on the use of 11C-Methionine in MM.

Furthemore, the use of 11C-Methionine (MET) in MM also seems to have a promising future. Lapa et al prospectively compared the sensitivity of 18FDG and MET for the detection of myeloma lesions in patients evaluated at baseline (n=11) or in relapse (n=32). PET-MET was able to detect LF and EMD in 6 and 2 more patients respectively than FDG-PET and also found more LF and EMD in 28 and 6 cases respectively. Thus, they demonstrated that MET uptake was correlated with bone marrow involvement, β2-microglobulin and free light chain levels, and appears to be a more accurate market of tumor burden compared to 18FDG (43). Another recent study confirmed the excellent sensitivity of MET for the assessment of MM tumor burden compared to 18FDG (44). Nevertheless, the prognostic value of this tracer need further investigation particularly in comparison with 18FDG, which is currently the strongest marker in this pathology.

  1. Lapa C, Knop S, Schreder M, Rudelius M, Knott M, Jörg G, et al. 11 C-Methionine-PET in Multiple Myeloma: Correlation with Clinical Parameters and Bone Marrow Involvement. Theranostics. 2016;6(2):254–61.
  2. Morales-Lozano MI, Viering O, Samnick S, Rodriguez-Otero P, Buck AK, Marcos-Jubilar M, et al. 18F-FDG and 11C-Methionine PET/CT in Newly Diagnosed Multiple Myeloma Patients: Comparison of Volume-Based PET Biomarkers. Cancers. 2020 Apr 23;12(4):1042.

2. More references are needed on the role of CXCR4; there are currently many free-standing sentences without references. It should also be mentioned that Pentixafor has been successfully applied in related blood cancers, i.e., PCNSL, MALT lymphoma, CLL and Morbus Waldenström.

Response:  Indeed, the text lacks reference about the role of CXCR4, two have been added:

  1. Ullah TR. The role of CXCR4 in multiple myeloma: Cells’ journey from bone marrow to beyond. Journal of Bone Oncology. 2019 Aug;17:100253.
  2. Peled A, Klein S, Beider K, Burger JA, Abraham M. Role of CXCL12 and CXCR4 in the pathogenesis of hematological malignancies. Cytokine. 2018 Sep;109:11–6.

And we also added a sentence on the interesting application of this tracer in other haematological diseases.

Interestingly, compared to 18FDG, 68Ga-Pentixafor also seems to be a promising imaging for assessment of other related blood cancers including MALT lymphoma (40), Waldenström macroglobulinemia, lymphoplasmacytic lymphoma (41) and primary central nervous system lymphoma (42).

  1. Haug AR, Leisser A, Wadsak W, Mitterhauser M, Pfaff S, Kropf S, et al. Prospective non-invasive evaluation of CXCR4 expression for the diagnosis of MALT lymphoma using [ 68 Ga]Ga-Pentixafor-PET/MRI. Theranostics. 2019;9(12):3653–8.
  2. Luo Y, Cao X, Pan Q, Li J, Feng J, Li F. 68 Ga-Pentixafor PET/CT for Imaging of Chemokine Receptor 4 Expression in Waldenström Macroglobulinemia/Lymphoplasmacytic Lymphoma: Comparison to 18 F-FDG PET/CT. J Nucl Med. 2019 Dec;60(12):1724–9.
  3. Ahn S-Y, Kwon SY, Jung S-H, Ahn J-S, Yoo SW, Min J-J, et al. Prognostic Significance of Interim 11C-Methionine PET/CT in Primary Central Nervous System Lymphoma. Clin Nucl Med. 2018 Aug;43(8):e259–64.

Reviewer 2 Report

In their review on 'FDG PET/CT , a promising exam for detecting high risk myeloma patients' authors have timely summarised the major findings of several studies in multiple myeloma patients examined with FDG PET. This review has the potential to serve as a basis for several future clinical trials.

Authors have structured their review article in an easy to follow and understand fashion. All major studies and their positive and negative points have been highlighted.

Some comments:

  1. Authors should consider providing a pictorial presentation of the stages where FDG PET/CT in inculcated in IMWG guidelines. For many readers who are unaware of the full guideline, such a graph will be extremely useful.
  2. Authors shold consider discussing pros and cons of SUVmax and othere semiquantitative tools for FDG uptake quantification in a separate paragraph. Is SUVmax the correct parameter or would authors rather go for other options e.g. SUVmean, SUVpeak etc.
  3. In table 1, acronym TEP-FDG abnormality is not described in text. Probably they meant PET, this needs to be corrected.
  4. Acronym PMD has not been defined in text
  5. Perspective: authors should also discuss on the use of PET/MRI in comparison to PET/CT in MM
  6. It will be a great help for the readers if authors could summarise the findings of FDG PET prospective clinical studies highlighting the major findings in a table.

Author Response

FDG-PET/CT, A PROMISING EXAM FOR DETECTING HIGH-RISK MYELOMA PATIENTS?

REPLY TO REVIEWER #2

We would like to thank the Reviewer #1 for all the valuable comments and for the careful review of the manuscript, which has helped to improve our work. Please find below a detailed point-by-point response to all comments (reviewers’ comments in black, our replies in blue, suggested text change in red).

  1. Authors should consider providing a pictorial presentation of the stages where FDG PET/CT in inculcated in IMWG guidelines. For many readers who are unaware of the full guideline, such a graph will be extremely useful.

Response: Thanks for this important suggestion, which makes the text clearer. We have added two figures (figures 4 and 6) summarizing the imaging algorithm for patients suspected of SMM/MM and SP.

  1. Authors shold consider discussing pros and cons of SUVmax and othere semiquantitative tools for FDG uptake quantification in a separate paragraph. Is SUVmax the correct parameter or would authors rather go for other options e.g. SUVmean, SUVpeak etc.

Response: Thanks for this remark. Several publications, in particular by our team (Carlier T, Bailly C. State-Of-The-Art and Recent Advances in Quantification for Therapeutic Follow-Up in Oncology Using PET. Front Med [Internet]. 2015 Mar 23 [cited 2020 May 25];2) have described in a general way the advantages and disadvantages of semi-quantitative FDG-PET/CT measurements, which are not specific to MM evaluation. However, we did find it interesting to add a sentence explaining this, with the associated reference: This publication by the Bologna group is the first that demonstrates the prognostic interest of the SUVmax in this indication. While many publications have already demonstrated the limitations of this quantitative parameter, it is useful for its easy use in routine practice (21).

  1. In table 1, acronym TEP-FDG abnormality is not described in text. Probably they meant PET, this needs to be corrected.

Response: Indeed, a mistake was made: TEP-FDG is FDG-PET/CT.

  1. Acronym PMD has not been defined in text.

Response: the definition has been added: para-medullary disease (PMD)

  1. Perspective: authors should also discuss on the use of PET/MRI in comparison to PET/CT in MM.

Response: Thank for this suggestion. Currently, there is few reference to the use of PET-MRI in the MM, but this technique could be developed in this indication, that is why it seems important to mention it.

In addition to the use of PET/CT, the emergence of PET/MRI could be an interesting way for exploring MM and others plasma cell dyscrasias, with the ability to image the whole body using both, PET and MRI by combining the excellent performances of these two exams, notably in high-risk patients, for initial staging, to assess treatment response and at relapse (45).

  1. Shah SN, Oldan JD. PET/MR Imaging of Multiple Myeloma. Magnetic Resonance Imaging Clinics of North America. 2017 May;25(2):351–65.

  1. It will be a great help for the readers if authors could summarise the findings of FDG PET prospective clinical studies highlighting the major findings in a table.

Response: Thank for this suggestion. For this purpose, we have clarified and reorganized Table 1, which summarizes the definition and prognosis of PET parameters.
